# Influence of Exogenous Neuropeptides on the Astrocyte Response Under Conditions of Continuous and Cyclic Hypoxia and Red Blood Cell Lysate

**DOI:** 10.3390/ijms26093953

**Published:** 2025-04-22

**Authors:** Klaudyna Kojder, Magdalena Gąssowska-Dobrowolska, Wojciech Żwierełło, Patrycja Kłos, Katarzyna Piotrowska, Agata Wszołek, Agnieszka Maruszewska, Izabela Gutowska, Dariusz Chlubek, Irena Baranowska-Bosiacka

**Affiliations:** 1Department of Anaesthesiology and Intensive Care, Pomeranian Medical University in Szczecin, Rybacka 1, 70-204 Szczecin, Poland; 2Department of Cellular Signalling, Mossakowski Medical Research Institute, Polish Academy of Sciences, Pawińskiego 5, 02-106 Warsaw, Poland; mgassowska@imdik.pan.pl; 3Department of Medical Chemistry, Pomeranian Medical University, Powstańców Wlkp. 72, 70-111 Szczecin, Poland; wojciech.zwierello@pum.edu.pl (W.Ż.); izabela.gutowska@pum.edu.pl (I.G.); 4Department of Biochemistry and Medical Chemistry, Pomeranian Medical University, Powstańców Wlkp. 72, 70-111 Szczecin, Poland; patrycja.klos@pum.edu.pl (P.K.); dchlubek@pum.edu.pl (D.C.); irena.baranowska.bosiacka@pum.edu.pl (I.B.-B.); 5Department of Physiology, Pomeranian Medical University, Powstańców Wlkp. 72, 70-111 Szczecin, Poland; katarzyna.piotrowska@pum.edu.pl; 6Institute of Biology, University of Szczecin, Wąska 13, 71-415 Szczecin, Poland; agata.wszolek@usz.edu.pl (A.W.); agnieszka.maruszewska@usz.edu.pl (A.M.)

**Keywords:** cerebrolysin, stroke, subarachnoidale hemorrhage, traumatic brain injury, acute brain injury, U87MG human brain cancer, cyclooxygenase 1 (COX1), cyclooxygenase 2 (COX2), prostaglandine E2 (PGE2), thromboxane B2 (TXB2), cytokines (IL-8, IL-1β, IL-6, IL-10), chemokines (CCL5/RANTES, CXCL9/MIG, CCL2/MCP-1, and CXCL10/IP-10)

## Abstract

Acute brain injury includes different pathologies: stroke, traumatic injury, subarachnoidale haemorhhage. In the pathophysiology of acute brain injury, secondary injury with hyperactivation of glia plays a crucial role. Activated glial cells induce prolonged inflammation that impacts the recovery and further cognitive functions of patients. In our study, we have examined the neuroprotective impact of exogenous neuropeptides—Cerebrolysin on astrocytes under different conditions. In a model that simulates central nervous system damage associated with brain injury, stroke, and subarachnoid hemorrhage, the U87MG human brain cancer (glioblastoma astrocytoma like) cells were treated with Cerebrolysin and exposed to conditions of continuous and cyclic hypoxia and red blood cell lysate overload. The activity and expression of cyclooxygenases COX-1 and COX-2 and on cytokines (IL-8, IL-1β, IL-6, IL-10) and chemokines (CCL5/RANTES, CXCL9/MIG, CCL2/MCP-1, and CXCL10/IP-10) concentration were assessed. Cerebrolysin lowers IL-1β and IL-6 and increases IL-10 under all conditions. Cerebrolysin may exhibit a neuroimmunotrophic function, reducing inflammation under conditions that replicate traumatic brain injury and hemorrhagic insults to the central nervous system. By modulating both pro-inflammatory and anti-inflammatory cytokines, Cerebrolysin can help create a more balanced immune response conducive to tissue repair and reduced secondary damage. Its ability to lower harmful mediators like IL-1β and IL-6 while enhancing protective factors such as IL-10 suggests a promising therapeutic strategy in stroke, traumatic brain injury, and subarachnoid hemorrhage. Alongside other mechanisms such as neurotrophic factor enhancement and glial cell regulation, this cytokine modulation underscores the therapeutic potential of Cerebrolysin in a variety of central nervous system disorders.

## 1. Introduction

Acute brain injury (ABI) accompanied by blood extravasation, whether it be hemorrhagic stroke, subarachnoid hemorrhage, or traumatic brain injury, is a sudden event that can lead to ischemia, blood–brain barrier leakage, edema, and escalating inflammation. Early edema, disruption of the blood–brain barrier, extravasated morphotic elements, and inflammation may help contain injury-related damage [1]. Over time, however, secondary or tertiary injury occurs with prolonged inflammation impeding tissue healing and regeneration. Diagnostic and therapeutic gestures, including neurosurgical decompression or neuroradiological intervention, and intensive care aimed at maintaining proper perfusion and intracranial pressure are therefore critical. Although mortality rates in ABI have improved, long-term outcomes, including return to pre-injury cognitive status and full social participation, remain unsatisfactory [2]. As a result, a multimodal approach incorporating both standard care and pharmacological interventions has been introduced [3].

Inflammation plays a pivotal role in the pathophysiological processes of ABI and influences patient prognosis [4]. It involves the activation of microglia, astrocytes, endothelial cells, and leukocytes leading to the release of pro-inflammatory cytokines and chemokines. Although the early inflammatory response can be beneficial, persistent inflammation may exacerbate tissue damage, contributing to secondary injury and complications such as motor or cognitive deficits. A thorough understanding of how inflammation is triggered, regulated, and sustained in brain injuries is thus essential for developing effective preventive and therapeutic strategies [5,6].

In the early phase of brain injury, transient global ischemia, oxidative stress, and inflammation lead to neuronal damage, brain edema, and blood–brain barrier disruption [5,6,7,8]. In subarachnoid hemorrhage, the inflammatory response has been associated with rats containing elevated levels of pro-inflammatory cytokines, including TNF-α and IL-1β, in both serum and brain tissue, as well as IL-1β, IL-4, IL-6, IL-8, IL-10, IL-15, IL-17, IL-18, TNF-α, MCP-1 (macrophage chemotactic protein-1), and VEGF (vascular endothelial growth factor) in cerebrospinal fluid during the first 7 days [9]. One study also reported higher IL-4 concentrations in the cerebrospinal fluid of patients who developed delayed ischemia, while IL-6 levels in serum on day 3 after subarachnoid hemorrhage correlated with clinical status at diagnosis and at 6 months [10].

Inflammation is a key factor driving tissue damage but also serves as an adaptive response aimed at promoting tissue repair [11]. Both acute and chronic inflammatory processes can cause additional harm and are linked to central nervous system pathologies [12]. Despite ongoing research, the pathogenesis of early brain injury, delayed ischemia, and accompanying cognitive dysfunction are not fully understood. Recent studies suggest that reactive gliosis—astroglial cell activation characterized by structural and functional changes—may be an important cellular mechanism behind complications after stroke and subarachnoidal hamorrhage (SAH) [13,14,15]. Peripheral immune cells and resident glial cells appear to play a central role in mounting the post-stroke immune response [4,16]. Reactive astrocytes can release cytokines, chemokines, growth factors, and neurotrophic factors that may either exacerbate or reduce neuroinflammation [17]. Elevated levels of interleukin-1 (IL-1), TNF-α, interleukin-6 (IL-6), interleukin-10 (IL-10), and transforming growth factor-β (TGF-β) have been noted, with IL-1 and TNF-α tending to worsen brain injury, whereas IL-6, IL-10, and TGF-β show neuroprotective properties [18].

Upon immune cell activation, the arachidonic acid (AA) cascade begins through phospholipase A_2_ (PLA_2_). AA is subsequently metabolized via the cyclooxygenase (COX) or lipoxygenase (LOX) pathways. The COX pathway converts AA into prostaglandins and thromboxane A_2_, implicated in microthrombi formation as a pathophysiological mechanism of delayed cerebral ischemia in intracerebral and subarachnoid hemorrhage (Ohashi et al., 2023 [10]). Cyclooxygenase 1 (COX-1) and cyclooxygenase 2 (COX-2)—both being prostaglandin-peroxide synthases—are crucial to inflammation. They catalyze the formation of prostaglandin H_2_ (PGH_2_) from AA, a precursor in prostanoid synthesis that includes prostaglandin E_2_ (PGE_2_) and thromboxane A_2_ (TXA_2_), which is converted into the more stable thromboxane B_2_ (TXB_2_) [19,20,21]. COX-1 was once regarded only as a constitutive enzyme with no role in inflammation. More recent data, however, indicate that COX-1 can be induced in certain tissues and participates in both the early phase of the pro-inflammatory response [20] and the progression of inflammation [22]. COX-2 is an inducible enzyme involved in various pathological processes [22], with pro-inflammatory cytokines recognized as factors triggering its production [23]. COX-2-catalyzed reactions represent the primary source of prostanoids such as PGE_2_ during the progression of inflammation [20].

Rapid diagnosis and intervention are key in managing intracerebral and subarachnoid hemorrhage. In recent years, classic neurosurgical procedures are being increasingly replaced by endovascular treatments that do not require craniectomy. In pursuit of improving both survival and better outcomes, including cognitive function, additional therapies—pharmacological among them—are under investigation. Cerebrolysin is one such agent used in intracerebral and subarachnoid hemorrhage. It is an exogenous peptide mixture derived from porcine brain tissue, administered as a slow infusion. In vitro and in vivo research have shown that Cerebrolysin can reduce astrogliosis and enhance neurogenesis in rats subjected to experimental traumatic brain injury [24]. In neuronal cultures derived from the cerebral cortex, Cerebrolysin has been observed to improve glutamine transporter function and to increase the activity of glutamine pathways [25]. In experimental models, Cerebrolysin has also been found to diminish inflammatory responses [26].

Inflammation induced by acute cerebral ischemia is a major factor in shaping stroke pathobiology and clinical course. The immune response begins locally in blocked and hypoperfused vessels and within ischemic brain parenchyma. Inflammatory mediators form in situ and then spread systemically. Because the precise mechanism of action of Cerebrolysin remains unclear, this study examined its effect on the activity and expression of cyclooxygenases COX-1 and COX-2, as well as on cytokines (IL-8, IL-1β, IL-6, IL-10) and chemokines (CCL5/RANTES, CXCL9/MIG, CCL2/MCP-1, and CXCL10/IP-10) levels in astrocytes under hypoxic conditions and in the presence of red blood cell lysate. This model simulates central nervous system injury associated with traumatic brain injury, stroke, and subarachnoid hemorrhage.

## 2. Results

### 2.1. Cell Culture Visualization

Under normoxic conditions, cells in the control group displayed standard growth. The systems containing Cerebrolysin or red blood cell (RBC) lysate, as well as the combination of both additives, did not differ from the control (Figure 1 below).

In continuous hypoxia with CoCl_2_, cells under all tested conditions tended to form clusters and spheroids, most notably in the control and Cerebrolysin wells. By contrast, cells incubated with RBC lysate or RBC lysate plus Cerebrolysin showed growth characteristics similar to those observed under normoxic conditions (Figure 2 below). The situation was quite different in the cyclic hypoxia conditions (Figure 3 below). 

### 2.2. The Effect of U87MG Cell Culture Exposure to CER, Red Blood Cells Lysate (RBC), and Red Blood Cells Lysate Together with (RBC + CER) on COX-1 and COX-2 Activity

#### 2.2.1. COX-1 and COX-2 Expression

Under normoxic conditions, cells incubated with RBC showed an increase in COX-1 and COX-2 expression compared to the control group (Figure 4A and Figure 5A). Cells incubated with CER showed reduced expression of both COX-1 and COX-2 compared to the control. In contrast, cells incubated with RBC + CER exhibited a marked decrease in COX-1 and COX-2 expression compared to cells incubated with RBC alone (Figure 4A and Figure 5A).

Similarly, under continuous and cyclic hypoxic conditions, COX-1 and COX-2 expression was upregulated in the RBC group compared to the control (Figure 4B,C and Figure 5B,C). In cells treated with CER, COX-1 and COX-2 expression was lower than in the control. In RBC + CER conditions, a decrease in both cyclooxygenases was observed relative to both the control and RBC groups (Figure 4 and Figure 5, Table 1 and Table 2).

#### 2.2.2. Effect of U87MG Cell Exposure to Cerebrolysin (CER), RBC Lysate (RBC), and RBC + CER on Prostaglandin E_2_ (PGE_2_) and Thromboxane B_2_ (TXB_2_) Levels

Under normoxic conditions, RBC-treated cells showed an increase in TXB_2_ compared with the control (approximately 10%, *p* = 0.0021). Cells incubated with CER exhibited a lower TXB_2_ level relative to the control (approximately 7%, *p* = 0.0332), as did RBC + CER-treated cells (approximately 22%, *p* < 0.0001) (Figure 6A). Under continuous and cyclic hypoxia, RBC-treated cells showed a significant rise in TXB_2_ (by about 40% and 45%, respectively, *p* < 0.0001) compared with the control. In contrast, CER-treated cells had reduced TXB_2_ relative to the control (by about 35%, *p* < 0.0001 under continuous hypoxia and 17%, *p* = 0.0332 under cyclic hypoxia). In RBC + CER-treated astrocytes, TXB_2_ was also decreased under both continuous and cyclic hypoxia (by approximately 25%, *p* < 0.0001 and 38%, *p* < 0.0001, respectively) (Figure 6B,C).

Under normoxic and continuous hypoxic conditions, RBC-treated cells demonstrated a statistically significant increase in PGE_2_ levels (by about 22%, *p* = 0.0021, and 38%, *p* = 0.0002, respectively) compared with the control. However, RBC + CER-treated cells had lower PGE_2_ levels compared with RBC alone (by about 15%, *p* = 0.0021, and about 17%, *p* = 0.0021, under normoxia and continuous hypoxia, respectively) (Figure 6D,E). Under cyclic hypoxia, RBC-treated cells displayed a significant elevation in PGE_2_ (approximately 42%, *p* < 0.0001) compared with the control, whereas RBC + CER-treated cells showed a decline in PGE_2_ of about 37% (*p* < 0.0001) relative to RBC alone. Additionally, CER-treated cells under cyclic hypoxia had lower PGE_2_ compared with the control (about 17%, *p* = 0.0332) (Figure 6F).

### 2.3. The Effect of U87MG Cell Culture Exposure to Cerebrolysine (CER), Red Blood Cells Lysate (RBC), and Red Blood Cells Lysate Together with Cerebrolysine (RBC + CER) on the Concentration of Cytokines: Il-1β, Il-6, Il-8, and Il-10

Figure 7 shows representative dot plots from the flow cytometry analysis. Each population of beads coated with antibodies specific to IL-1β, IL-6, IL-8, or IL-10, is distinguished by unique fluorescence intensity and stained with a PE-conjugated detection antibody. The PE fluorescence intensity of each complex reflects the concentration of the corresponding cytokine. Data for individual proteins were analyzed using CytExpert software (Beckman Coulter, version 2.6.0.105).

As shown in Figure 8, RBC exposure led to significant alterations in some cytokines. Under normoxic conditions, IL-6 decreased by about 68% (*p* < 0.0001) and IL-8 by about 48% (*p* = 0.0021) (Figure 8D,G). Under continuous hypoxia, IL-1β increased by about 522% (*p* < 0.0001), while IL-6 and IL-8 decreased by about 32% (*p* < 0.0001) and 61% (*p* < 0.0001), respectively (Figure 8B,E,H). Under cyclic hypoxia, IL-1β, IL-6, and IL-8 all declined (approximately 50%, 52%, and 56%, respectively; *p* = 0.0002 or *p* < 0.0001) (Figure 8C,F,I). RBC did not affect IL-10 under any of the tested conditions (Figure 8J–L).

Interestingly, incubation with CER alone significantly raised IL-1β by about 201% (*p* < 0.0001) under normoxia and 213% (*p* < 0.0001) under continuous hypoxia (Figure 8A,B). CER also increased IL-6 levels under normoxic, continuous, and cyclic hypoxic conditions by about 195% (*p* < 0.0001), 156% (*p* < 0.0001), and 9% (*p* < 0.0001), respectively (Figure 8D–F). A stimulatory effect of CER on IL-8 was noted under normoxia (108% increase, *p* = 0.0332) and cyclic hypoxia (52% increase, *p* < 0.0001) (Figure 8G,I), while under continuous hypoxia IL-8 decreased by about 5% (*p* < 0.0001) (Figure 8H). CER significantly elevated IL-10 by about 679% (*p* = 0.0021), but only under cyclic hypoxia (Figure 8L).

In RBC + CER-treated cells, CER markedly reduced IL-1β by about 67% (*p* < 0.0001) and IL-8 by about 21% (*p* < 0.0001) under continuous hypoxia (Figure 8B), and IL-6 and IL-8 by about 5% (*p* = 0.0332) and 9% (*p* = 0.0002), respectively, under cyclic hypoxia (Figure 8F,I). However, IL-1β increased by about 214% (*p* < 0.0001) (Figure 8C) and IL-6 and IL-8 both rose (by about 107%, *p* < 0.0001, and about 15%, *p* = 0.0021, respectively) (Figure 8D,G) under normoxia. RBC + CER did not significantly affect IL-1β levels under normoxia, IL-6 under continuous hypoxia, or IL-10 under any of the conditions tested (Figure 8A,E,J–L).

### 2.4. Effect of U87MG Cell Exposure to Cerebrolysin (CER), RBC Lysate (RBC), and RBC + CER on CCL5, CXCL9, CCL2, and CXCL10 Levels

Figure 9 shows representative dot plots from flow cytometry analysis. Each bead population, distinguished by different fluorescence intensity, was coated with specific antibodies for CCL5, CXCL9, CCL2, or CXCL10 and then stained with a PE-conjugated detection antibody. The PE fluorescence intensity of each complex indicates the concentration of the respective chemokine. Data were analyzed using CytExpert software (Beckman Coulter).

As shown in Figure 10, analysis of these inflammation-related chemokines revealed statistically significant changes following RBC exposure. Under normoxic, continuous, and cyclic hypoxic conditions, RBC-treated cells showed significantly increased CCL5 (*p* < 0.0001): approximately 55,085% under normoxia, 61,588% under continuous hypoxia, and 4560% under cyclic hypoxia (Figure 10A–C). CXCL9 also rose from zero in controls to 0.3939 (*p* = 0.0021), 0.709 (*p* < 0.0001), and 0.5253 (*p* < 0.0001), respectively (Figure 10D–F). For CCL2, a significant decrease (*p* < 0.0001) of about 35% was observed only under cyclic hypoxia (Figure 10G–I). RBC did not affect CXCL10 levels under any tested condition (Figure 10J–L).

Cerebrolysin (CER) alone did not alter CCL5 or CXCL9 under normoxia or hypoxia (Figure 10A–F). However, it significantly lowered CCL2 by about 78% (*p* = 0.0002) in normoxia and 45% (*p* < 0.0001) under cyclic hypoxia (Figure 10G,I). CXCL10 was reduced to zero (*p* < 0.0001) by CER under continuous hypoxia (Figure 10K).

In RBC + CER-treated cultures, CER significantly decreased CCL2 by about 22% (*p* = 0.0332) under normoxia (Figure 10G), as well as CXCL9 by 32% (*p* = 0.0332) and CXCL10 by 43% (*p* = 0.0332) under continuous hypoxia (Figure 10E,K). Under cyclic hypoxia, CCL5, CXCL9, and CCL2 were also reduced (by about 20%, *p* < 0.0001; 19%, *p* = 0.0332; 36%, *p* < 0.0001, respectively) (Figure 10C,F,I). In contrast, under normoxia, RBC + CER caused an approximately 12% increase in CCL5 (*p* = 0.0021) and 46% increase in CXCL9 (*p* = 0.0332) (Figure 10A,D). For the remaining chemokines, CER showed no significant effect in RBC-exposed cells (Figure 10B,H,J,L).

## 3. Discussion

Secondary inflammation processes play a critical role in the pathogenesis of acute and chronic nervous system injury. Current research efforts focus on ways to modulate this process, since elucidating its underlying mechanisms may lead to new therapeutic strategies. Cerebrolysin, a mixture of low-molecular-weight neuropeptides and free amino acids, is one potential agent for regulating inflammation and promoting neuroprotection, neuroplasticity, and neuronal regeneration [27,28]. This brain-specific, pleiotropic neuroprotective compound has been proposed as a promising option for stroke and traumatic brain injury [26,29,30,31].

Previous studies show that Cerebrolysin acts through multiple molecular pathways to enhance functional recovery following neurological diseases and injuries, including interactions with GluR1 and GABA RA/B receptors, regulation of intracellular signaling cascades (PI3K/Akt, NF-κB, JNK, and p38-MAPK), and alterations in molecular mediators relevant to trophic activity (NGF, BDNF, IGF-1), glucose transport (GLUT1), inflammation (TNF-α, IL-1β), and neurotransmission (cholinergic, glutamatergic, and GABAergic) [25,26,32,33,34]. Cerebrolysin’s neuroprotective and neurotrophic properties have been confirmed both in vitro and in animal models of ischemic stroke, as well as in clinical studies involving ischemic stroke patients and those with traumatic brain injury diagnosis [25,26,31,35,36]. These properties encompass anti-apoptotic activity, mitigation of glutamate excitotoxicity, reduction of free oxygen radicals, and modulation of microglial activation and neuroinflammation [25,29,30,31,34,37,38,39,40,41,42,43].

In strokes, reactive astrocytes commonly release pro-inflammatory cytokines that can exacerbate neuronal damage. Cerebrolysin can moderate this inflammatory response. Some studies indicate that it significantly improves functional recovery in rat models with mild traumatic brain injury by reducing astrogliosis and axonal damage, enhancing neurogenesis, and lowering blood–brain barrier (BBB) disruption [31]. In a glutamate toxicity model, Cerebrolysin also decreases IL-1β levels and raises IL-10 levels, thereby mitigating glutamate-mediated neuroinflammation [25]. IL-10 plays a critical protective role by inhibiting pro-inflammatory cytokine release, thus helping suppress both immunoproliferative and inflammatory processes [44]. Because an imbalance between pro-inflammatory and anti-inflammatory cytokines can significantly affect stroke outcomes, an IL-10 increase driven by Cerebrolysin may foster a more balanced immune response and support tissue repair. This mechanism may also limit excessive inflammation and protect brain tissue in cases of injury. After traumatic brain injury (TBI), Cerebrolysin enhances neurological scores, lowers cerebral edema, improves BBB permeability, and reduces inflammatory cytokines such as TNF-α, IL-1β, IL-6, and NF-κB [26]. Parallel findings have been reported in TBI patients, where Cerebrolysin decreases TNF-α, IL-1β, IL-6, and NF-κB. Moreover, Cerebrolysin can downregulate inflammatory responses in intracranial hemorrhage. In murine intracerebral hemorrhage models, Cerebrolysin-treated animals exhibited significant reductions in IL-1β, IL-6, and TNF-α, as well as in aquaporin-4 (AQP4)—a main mediator of vasogenic edema. A decrease in cytotoxic astrocyte edema was also noted [45].

Despite these benefits, the effect of Cerebrolysin in hemorrhagic stroke patients remains contested [29,46]. In a randomized, placebo-controlled, double-blind pilot trial of aneurysmal subarachnoid hemorrhage (SAH), adding Cerebrolysin to standard-of-care therapy proved safe, well tolerated, and feasible but did not notably improve six-month global functional outcomes [43]. Although data from intracranial hemorrhage (ICH) models are limited, Cerebrolysin’s potential to modulate cytokine expression—affecting both pro and anti-inflammatory factors—suggests it could help promote a more balanced inflammatory state and better tissue repair. The specific impact on cytokine levels may vary by the neurological condition and stage of disease or injury.

Few investigations have examined astrocytes’ role in stroke, and their function remains incompletely defined. Astrocytes can be beneficial or harmful, depending on the stroke subtype and timing. For instance, aneurysmal SAH has distinct pathophysiological mechanisms compared to ischemic stroke. This presents a challenge in harnessing astrocytes’ protective roles while limiting their detrimental effects. A recent TBI study found that astrocytes become reactive not just near the primary lesion but also in distant brain areas. Homeostatic astroglial proteins activate close to the lesion, and pro-inflammatory genes are overexpressed, influencing long-term recovery and cognitive function. While the initial glial scar helps prevent extensive immune cell infiltration, it eventually hinders neuronal reconnections. Secondary processes such as edema, blood–brain barrier disruption, and neuroinflammation can lead to a concept often termed tertiary brain damage [47,48].

Cytokine upregulation in the brain after various injuries—including stroke—occur not only in immune cells but also in resident brain cells such as glia and neurons. Key inflammatory cytokines in stroke include interleukin-1 (IL-1), TNF-α, interleukin-6 (IL-6), interleukin-10 (IL-10), and transforming growth factor-β (TGF-β). Of these, IL-1 and TNF-α typically exacerbate brain injury, whereas IL-6, IL-10, and TGF-β often provide neuroprotective effects [49]. Arachidonic acid metabolism involving COX-1 and COX-2 is also essential in brain pathology. Mouse studies show that certain pathological conditions either upregulate or downregulate COX-1 and COX-2 in the injured brain [50]. IL-8, a chemokine activated through CXCR1 and CXCR2, is produced by cells such as monocytes, leukocytes, and epithelial cells [51]. In a study of severe TBI patients, IL-8 concentrations were higher in the cerebrospinal fluid than in plasma, suggesting intrathecal production, which correlated with nerve growth factor (NGF) production and BBB impairment [52]. In another investigation of plasma cytokines in TBI, IL-6, IL-8, IL-10, IL-12p70, and TNF-α were linked to TBI diagnosis, while IL-10, IL-8, and IL-6 were associated with mortality. In contrast, IL-1β, TNF-α, and IL-12p70 levels did not differ between survivors and non-survivors, and IL-1β did not rise in TBI patients [53]. Similar work also highlights IL-6, IL-8, and IL-10 as possible early predictors of poor outcomes in TBI patients, with IL-8 notably correlating with mortality [54,55]. Large-scale research involving 166 participants linked elevated IL-8, IL-10, and TNF-α to worse prognoses at six months. However, studies on cytokine involvement in SAH have yielded mixed results. Rasmussen et al. (2019) found no correlation between plasma concentrations of IL-6, IL-8, IL-10, ICAM-1, VCAM-1, IFNγ, and TNF-α and outcomes like delayed cerebral ischemia (DCI) or infarction [56]. Conversely, Luo et al. observed significantly higher levels of IL-6, IL-10, IL-8, IL-2, and TNF-α in patients with poorer aneurysmal SAH outcomes (*n* = 200) [17]. Other research suggests that IL-6 and IL-1β correlate with vasospasm and IL-1β correlates with worse outcomes in mild TBI [57,58]. Additional animal studies linked IL-1β to blood–brain barrier disruption, cerebral edema, and immune cell activation [59,60]. Similarly, a study of 81 SAH patients found elevated levels of IL-1β, IL-18, and TNF-α in cerebrospinal fluid [61]. IL-1β has been shown to worsen post-stroke inflammation and blood–brain barrier dysfunction, facilitating neuronal apoptosis. IL-8 accelerates inflammation and tissue damage, partly by promoting neutrophil activity. Meanwhile, IL-10, detected on microglia, and its receptor on astrocytes in experimental stroke models, enhance neuronal survival. Low IL-10 levels correlate with poorer outcomes [62].

Our investigation into Cerebrolysin’s impact on cyclooxygenase expression and activity, as well as on cytokine and chemokine levels in astrocytes under hypoxia and in the presence of red blood cell lysate, mimicking traumatic brain injury, hemorrhagic stroke, and subarachnoid hemorrhage, revealed beneficial effects on glial cells. These findings align with clinical research on TBI [63,64,65]. For other acute brain injuries like SAH, data remain inconclusive, underscoring the need for additional molecular and clinical studies [43,66]. While no current data examine Cerebrolysin in a subarachnoid hemorrhage model similar to hemorrhagic stroke, we aimed to assess astrocyte responses under these conditions. A central question is why some clinical results on Cerebrolysin used in TBI, SAH, or stroke show less robust statistical significance than those seen in TBI alone. The heterogeneity of these disorders and variations in patient populations may play a role, along with external factors that could hinder Cerebrolysin’s beneficial mechanisms. By investigating astrocyte responses under cyclic or continuous hypoxia and hemorrhagic conditions, we observed that Cerebrolysin reduced cell clustering and spheroidal deformation—key features induced by hypoxia and RBC lysate exposure—thus mitigating effects akin to those seen in hemorrhagic stroke. Few, if any, published studies have explored cyclic and continuous hypoxia combined with hemorrhage-related conditions to examine their collective impact on astrocytic inflammation. One in vitro mouse brain astrocyte model (CRL-2541) preincubated with hemolysate demonstrated elevated COX-2 [45]. Indeed, inflammation is central to secondary brain injury after hemorrhagic stroke (ICH and SAH) [67]. Hemolysate produced by intracranial hemorrhage amplifies brain edema via an inflammatory response, while lysis of red blood cells accelerates edema and BBB permeability [68,69]. It has long been used to simulate SAH in animal models and is implicated in secondary damage in ICH [70,71]. We found that hemolysate enhanced COX-1 and COX-2 expression, intensifying astrocytic inflammation. Although our data support this mechanism, the in vitro model does not perfectly replicate in vivo hemorrhage, and the precise neurotoxic effect of hemolysate requires further study.

Our results suggest that Cerebrolysin exerts a protective influence, as seen in the downregulation of COX-1 and COX-2 (lower TBX_2_ and PGE_2_) following RBC exposure under both continuous and cyclic hypoxia, along with reduced expression of some cytokines (IL-1β and IL-8 under continuous hypoxia; IL-6 and IL-8 under cyclic hypoxia) and chemokines (CXCL9 and CXCL10 under continuous hypoxia; CCL5, CXCL9, and CCL2 under cyclic hypoxia). This pattern supports a more controlled astrocytic response. Interestingly, Cerebrolysin by itself boosted the production of various cytokines (IL-1β, IL-6, and IL-8 under normoxia; IL-1β and IL-6 under continuous hypoxia; IL-1β, IL-6, IL-8, and IL-10 under cyclic hypoxia). Rather than completely blocking the inflammatory response, Cerebrolysin appears to modulate and temper it, which helps preserve BBB integrity in primary injury and curb astrogliosis in secondary processes. Our data also reveal that Cerebrolysin’s effects were more pronounced under hypoxic conditions, whether cyclic or continuous, than in astrocytes additionally exposed to RBC. This finding may explain why hemorrhage severity or residual intracranial blood correlates with poorer outcomes clinically, as direct toxicity from red blood cells may limit Cerebrolysin’s beneficial effects.

## 4. Materials and Methods

### 4.1. Reagents

The U87MG human brain cancer (glioblastoma astrocytoma-like) cell line was obtained from the American Type Culture Collection (ATCC, Rockville, MD, USA). EMEM medium, glutamine, amino acids, sodium pyruvate, antibiotics (penicillin and streptomycin), phosphate-buffered saline (PBS), 0.1 M cobalt chloride solution, and 0.25% Trypsin-EDTA were purchased from Sigma-Aldrich (Poznań, Poland). Fetal bovine serum (FBS) was obtained from Gibco (Paisley, UK). Cerebrolysin was provided by EVER Neuro Pharma (Unterach, Austria).

### 4.2. Cell Culture and Treatment

Experiments were performed on U87MG glioblastoma astrocytoma-like cells grown in Minimum Essential Medium Eagle (EMEM; Sigma-Aldrich, Saint Louis, MO, USA). The medium contained 2 mM glutamine (Sigma-Aldrich), 1% non-essential amino acids (Sigma-Aldrich), 1 mM sodium pyruvate (Sigma-Aldrich), and antibiotics (100 IU/mL penicillin and 10 µg/mL streptomycin; Sigma Aldrich). It was supplemented with 10% thermally inactivated FBS (Gibco, Waltham, MA USA). Cultures were maintained in a humidified incubator at 37 °C with 5% CO_2_.

Cells were seeded in six-well plates (Nest, Scientific Biotechnology, Wuxi, China) at 20,000 cells/cm^2^ and cultured for 72 h until 70–80% confluence was reached. The spent medium was then removed, and the cells were gently rinsed three times with 37 °C PBS (Sigma-Aldrich, Saint Louis, MO, USA). Test media were subsequently added, and cultures continued for 48 h. Four experimental groups were established: control (complete standard medium), Cerebrolysin-treated (125 µg/mL), red blood cell lysate-treated (10%), and combined treatment with Cerebrolysin (125 µg/mL) plus red blood cell lysate (10%).

Cells were exposed to normoxia (37 °C, 95% humidity, 5% CO_2_), continuous hypoxia (+200 µM CoCl_2_, 37 °C, 95% humidity, 5% CO_2_), or cyclic hypoxia (1% O_2_ for 4 h → 21% O_2_ for 4 h → 1% O_2_ for 12 h → 21% O_2_ for 4 h → 1% O_2_ for 4 h → 21% O_2_ for 4 h → 1% O_2_ for 12 h). Subsequent steps varied according to the objectives of each culture experiment.

### 4.3. Determination of COX-1 and COX-2 Activity

COX-1 and COX-2 activity were quantified by measuring prostaglandin E_2_ (PGE_2_) and thromboxane B_2_ (TXB_2_) in cell lysates and culture supernatants. Cell lysates were prepared using RIPA buffer (50 mM Tris, 0.9% NaCl, 0.1% SDS, pH 7.3). Protein content was standardized using the BCA assay. Culture supernatants were analyzed directly and normalized to total protein concentration. All measurements were performed in duplicate to improve data reliability.

#### 4.3.1. Imaging of COX-1and COX-2 Expression

COX-1 and COX-2 levels were detected by fluorescence microscopy. U87MG cells (5 × 10^5^) were treated with 125 µg/mL Cerebrolysin, 10% RBC lysate, or both (CER + RBC). Control cells were cultured in standard medium without additives. Incubations were under normoxic (37 °C, 95% humidity, 5% CO_2_), continuous hypoxic (+200 mM CoCl_2_), or cyclic hypoxic conditions (1% O_2_ for 4 h → 21% O_2_ for 4 h → 1% O_2_ for 12 h → etc.) for 48 h. Following incubation, cells were washed in PBS and used for confocal microscopy to visualize COX-1 and COX-2 proteins. Cells were cultured on glass coverslips in six-well plates under standard conditions, washed with PBS, fixed for 15 min at room temperature with 4% buffered formalin, and then washed again. Permeabilization was performed with 0.5% Triton X-100 in PBS. After additional washing, cells were incubated for 1 h at room temperature with mouse anti-COX-1 and anti-COX-2 antibodies (1:50; Santa Cruz Biotechnology, Dallas, TX, USA). They were then rinsed and further incubated for 45 min at room temperature with anti-mouse IgG FITC-conjugated secondary antibodies (1:80; Sigma-Aldrich, Poznań, Poland). After another wash in PBS, the nuclei were stained with DAPI for 15 min at room temperature. Samples were examined on an Olympus FV1000 confocal microscope (IX81 inverted platform), using two-channel acquisition and sequential scanning to separately detect DAPI and FITC fluorescence. Enzyme expression was visible as green fluorescence. Merged fluorescence images were then prepared. Experiments were performed as six separate assays for each COX (each assay in three replicates).

#### 4.3.2. Determination of PGE_2_ and TXA_2_ Level

PGE_2_ and TXA_2_ concentrations were measured using FineTest^®^ ELISA kits (Wuhan Fine Biotech Co. Ltd., Wuhan, China) catalog numbers EU2554 for PGE_2_ and EU2638 for TXA_2_) based on a competitive ELISA method. The PGE_2_ or TXA_2_ present in samples competed with the labeled antibodies for binding sites on the microtiter plates. An enzymatic reaction with TMB generated a color that was read at 450 nm on a microplate reader. Analyte concentrations were derived from standard curves prepared for each assay set.

To reduce background, plates were washed with a buffer before adding 50 µL of samples (lysates or medium) or calibration standards. A biotinylated antibody specific to the analyte of interest was then added, and the plates were incubated for 45 min at 37 °C. After thorough washing, 100 µL of HRP-Streptavidin conjugate was applied, and incubation continued for 30 min at 37 °C. Each standard curve was generated separately for each run and each sample was examined in duplicate to minimize technical errors.

### 4.4. Determination of Cytokines and Chemokines Concentration

Concentrations of IL-8, IL-1β, IL-6, IL-10, CCL5/RANTES, CXCL9/MIG, CCL2/MCP-1, and CXCL10/IP-10 were evaluated by flow cytometry using the BD Cytometric Bead Array (CBA) Human Inflammatory Cytokines Kit and the BD CBA Human Chemokine Kit (Becton Dickinson, Franklin Lakes, NJ, USA). Culture supernatants were collected from cells treated with blood hemolysate and/or Cerebrolysin under normoxic, hypoxic, and cyclic hypoxic conditions. Control samples included media from untreated cells and various background controls (serum-free media, serum-supplemented media, and media containing hemolysate and/or Cerebrolysin).

Cytokine and chemokine concentrations were determined from standard curves generated using reference materials provided with the kits. Samples or standards were incubated for three hours in the dark with the Cytokine/Chemokine Capture Beads Mix and the Human Inflammatory Cytokine PE Detection Reagent. After this step, they were washed, centrifuged, and resuspended in Wash Buffer. Yellow–orange fluorescence was detected at 585/42 nm (λ_ex = 488 nm) using a CytoFlex flow cytometer (Beckman Coulter, Brea, CA, USA). Each bead population was distinguished by its far-red fluorescence (780/60 nm, λ_ex = 488 nm).

### 4.5. Statistical Analysis

All data were analyzed using GraphPad Prism version 8 (GraphPad Software, San Diego, CA, USA). One-way analysis of variance (ANOVA) followed by the Newman–Keuls post-hoc test was employed. Results were expressed as the mean ± SEM, and *p* < 0.05 was considered statistically significant.

## 5. Conclusions

In conclusion, these in vitro findings indicate that Cerebrolysin may exhibit a neuroimmunotrophic function, reducing inflammation under conditions that replicate traumatic brain injury and hemorrhagic insults to the central nervous system. By modulating both pro-inflammatory and anti-inflammatory cytokines, Cerebrolysin can help create a more balanced immune response conducive to tissue repair and reduced secondary damage. Its ability to lower harmful mediators like IL-1β and IL-6 while enhancing protective factors such as IL-10 suggests a promising therapeutic strategy in stroke, traumatic brain injury, and subarachnoid hemorrhage. Alongside other mechanisms—such as neurotrophic factor enhancement and glial cell regulation—this cytokine modulation underscores the therapeutic potential of Cerebrolysin in a variety of central nervous system disorders.

## Figures and Tables

**Figure 1 ijms-26-03953-f001:**
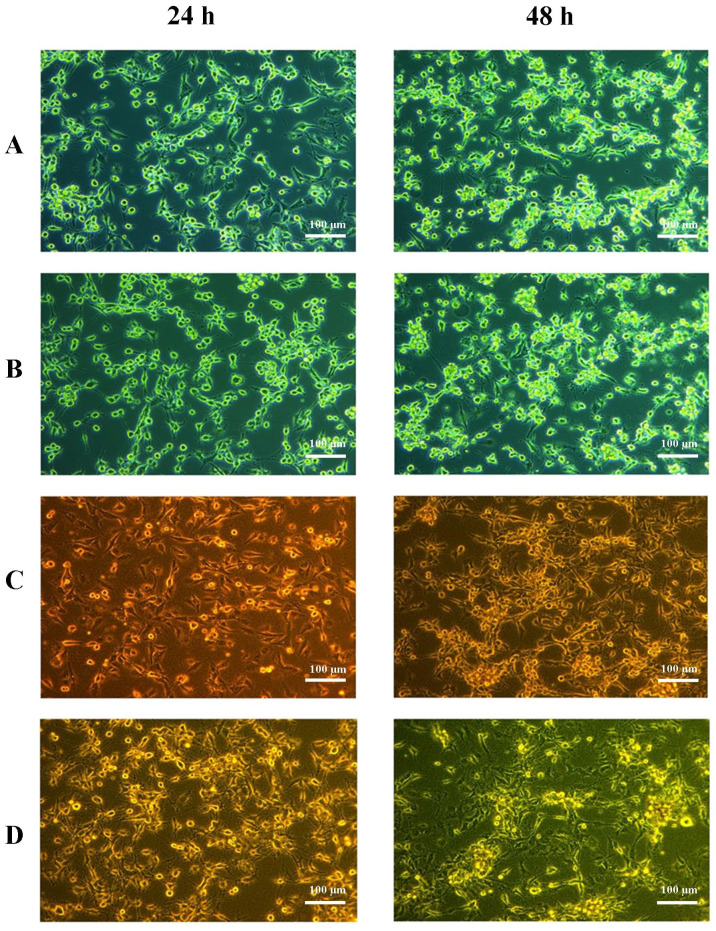
Imaging of U87MG cultures under normoxic conditions. Cells were cultured in supplemented EMEM medium (control) for 24 h and 48 h (**A**), with 125 µg/mL Cerebrolysin for 24 h and 48 h (**B**), with 10% RBC lysate for 24 h and 48 h (**C**), or with both 125 µg/mL Cerebrolysin and 10% RBC lysate for 24 h and 48 h (**D**). Experiments were performed as six separate assays (each assay in three replicates).

**Figure 2 ijms-26-03953-f002:**
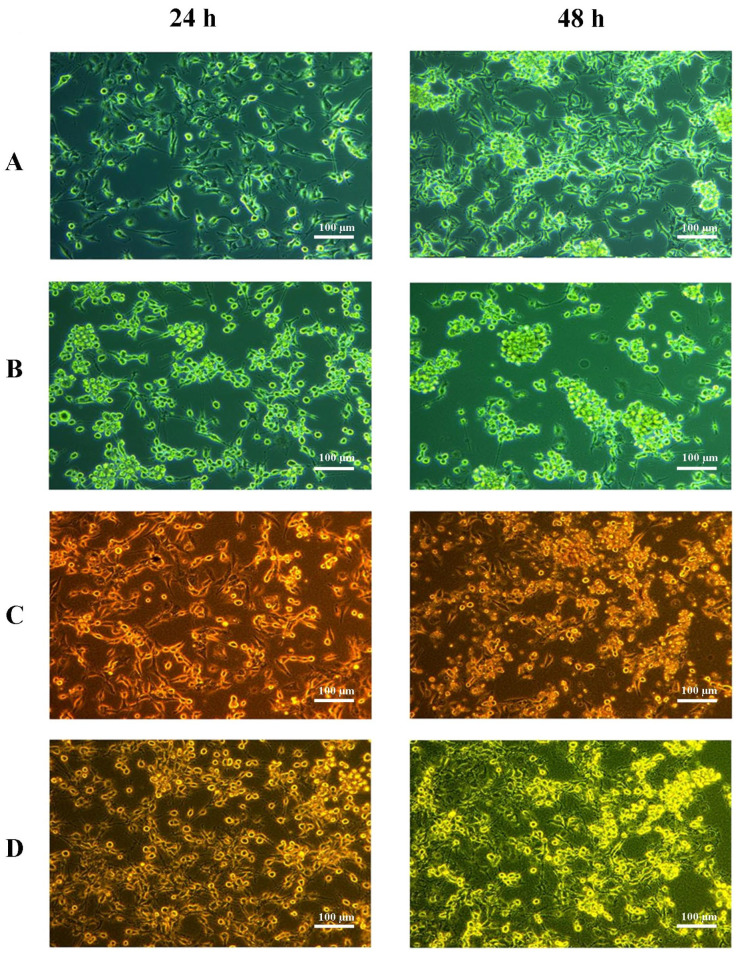
Imaging of U87MG cultures under continuous hypoxic conditions. Cells were cultured in supplemented EMEM medium containing 200 mM CoCl_2_ (control) for 24 h and 48 h (**A**), with 125 µg/mL Cerebrolysin plus 200 mM CoCl_2_ for 24 h and 48 h (**B**), with 10% RBC lysate plus 200 mM CoCl_2_ for 24 h and 48 h (**C**), or with 125 µg/mL Cerebrolysin, 10% RBC lysate, and 200 mM CoCl_2_ for 24 h and 48 h (**D**). Experiments were performed as six separate assays (each assay in three replicates).

**Figure 3 ijms-26-03953-f003:**
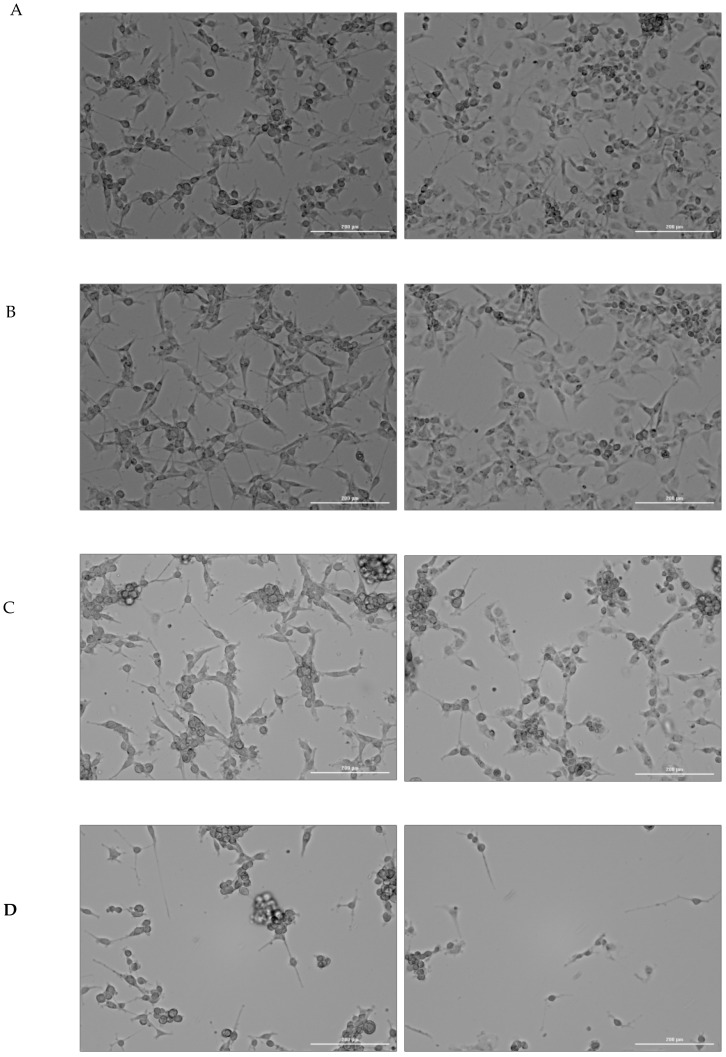
Imaging of U87MG cultures under cyclic hypoxic conditions. Time-lapse imaging of U87MG cells under cyclic hypoxia. Images were captured at the end of the first (panel I) and shortly before the last (panel II) 12-h hypoxia (1%) period. RBC-treated cells (**A**) and cells treated with RBC plus Cerebrolysin (**B**) initially form aggregates (panel I) and then disaggregate, growing as a monolayer (panel II). Cells treated with Cerebrolysin alone (**C**) assemble into spheres (panel I) and remain in that form until the end of the final reoxygenation period (panel II). Control cells (**D**), cultured in supplemented EMEM without Cerebrolysin or RBC, develop spheroids (panel I) and detach from the plate before the last 12 h hypoxic period (panel II). Cells were imaged in bright-field mode using a 10× objective. Scale bar—200 µm.

**Figure 4 ijms-26-03953-f004:**
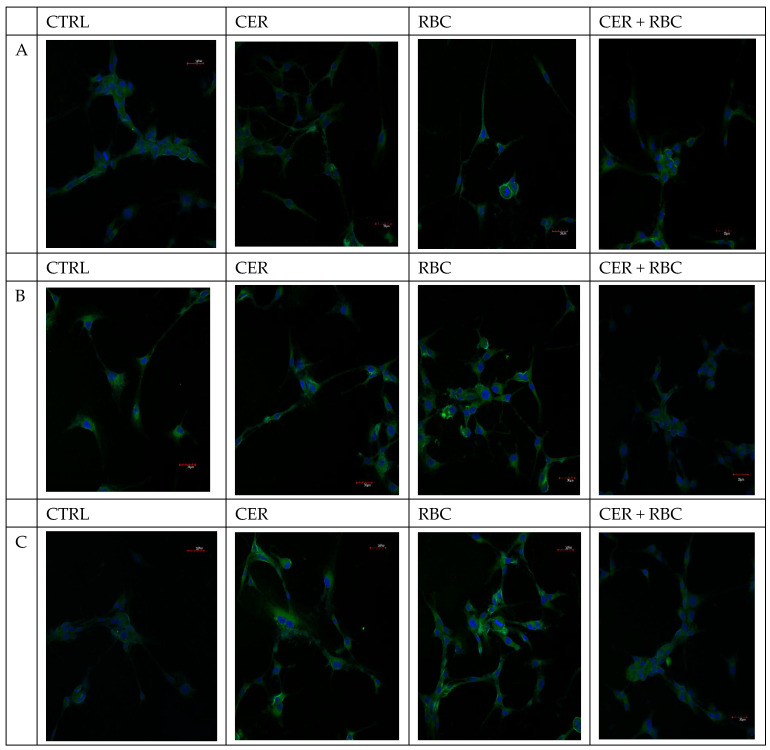
Effect of U87MG cell exposure to cerebrolysin (CER), RBC lysate (RBC), and RBC + CER on COX-1 expression. COX-1 levels were detected by fluorescence microscopy. Cells were treated with 125 µg/mL Cerebrolysin, 10% RBC lysate, or both (CER + RBC). Control cells were cultured in standard medium without additives. Incubations were under normoxic (37 °C, 95% humidity, 5% CO_2_) (**A**), (**B**) continuous hypoxic (+200 mM CoCl_2_), or (**C**) cyclic hypoxic conditions (1% O_2_ for 4 h → 21% O_2_ for 4 h → 1% O_2_ for 12 h → etc.) for 48 h. Immunofluorescence used a mouse anti-COX-1 antibody (1 h at RT) followed by a FITC-conjugated secondary antibody (45 min at RT). Enzyme expression is visible as green fluorescence (red arrows indicate upregulation vs. the control; white arrows indicate downregulation vs. the control; blue arrows indicate downregulation vs. RBC). Nuclei were stained with DAPI. Experiments were performed as six separate assays (each assay in three replicates). Scale bar = 30 µm.

**Figure 5 ijms-26-03953-f005:**
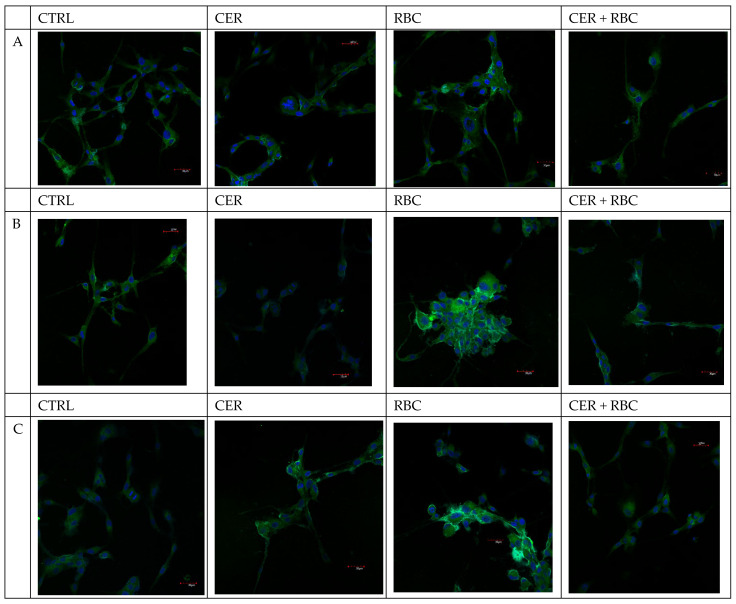
Effect of U87MG cell exposure to cerebrolysin (CER), RBC lysate (RBC), and RBC + CER on COX-2 expression. COX-2 levels were assessed by fluorescence microscopy. Cells were treated with 125 µg/mL cerebrolysin, 10% RBC lysate, or both (CER + RBC). Control cells were cultured in standard medium. Incubations were under (**A**) normoxic (37 °C, 95% humidity, 5% CO_2_), (**B**) continuous hypoxic (+200 mM CoCl_2_), or (**C**) cyclic hypoxic conditions (1% O_2_ for 4 h → 21% O_2_ for 4 h → 1% O_2_ for 12 h → etc.) for 48 h. Immunofluorescence used a mouse anti-COX-2 antibody (1 h at RT), followed by a FITC-conjugated secondary antibody (45 min at RT). Enzyme expression is visible as green fluorescence (red arrows indicate upregulation vs. the control; white arrows indicate downregulation vs. the control; blue arrows indicate downregulation vs. RBC). Nuclei were stained with DAPI. Experiments were performed as six separate assays (each assay in three replicates). Scale bar = 30 µm.

**Figure 6 ijms-26-03953-f006:**
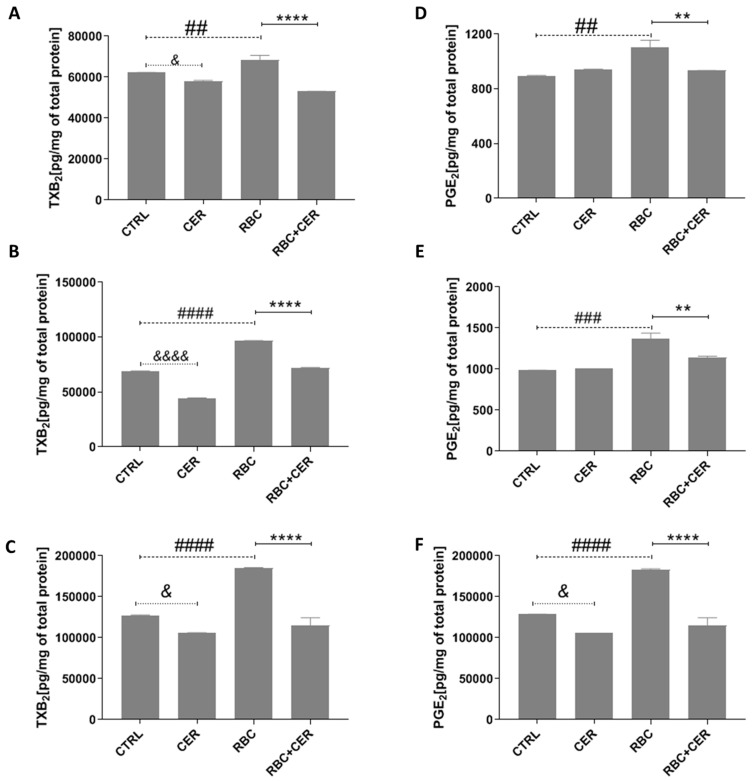
Effect of U87MG cell exposure to cerebrolysin (CER), RBC lysate (RBC), and RBC + CER on Prostaglandin E_2_ (PGE_2_) and Thromboxane B_2_ (TXB_2_). Cells were treated with 125 µg/mL cerebrolysin, 10% RBC lysate, or both (RBC + CER). Control cells were cultured in standard medium without additives. Cultures were maintained under (**A**,**D**) normoxia (37 °C, 95% humidity, 5% CO_2_), (**B**,**E**) continuous hypoxia (+200 mM CoCl_2_, 37° C, 95% humidity, 5% CO_2_), or (**C**,**F**) cyclic hypoxia (1% O_2_ for 4 h → 21% O_2_ for 4 h → 1% O_2_ for 12 h → 21% O_2_ for 4 h → 1% O_2_ for 4 h → 21% O_2_ for 4 h → 1% O_2_ for 12 h). Cells were cultured for 48 h. Data represent mean values ± SEM from three independent experiments (*n* = 3). (**A**) & *p* = 0.0332 vs. CTRL, ## *p* = 0.0021 vs. CTRL, **** *p* < 0.0001 vs. RBC; (**B**) &&&& *p* < 0.0001 vs. CTRL, #### *p* < 0.0001 vs. CTRL, **** *p* < 0.0001 vs. RBC; (**C**) & *p* = 0.0332 vs. CTRL, #### *p* < 0.0001 vs. CTRL, **** *p* < 0.0001 vs. RBC; (**D**) ## *p* = 0.0021 vs. CTRL, ** *p* = 0.0021 vs. RBC; (**E**) ### *p* = 0.0002 vs. CTRL, ** *p* = 0.0021 vs. RBC; (**F**) & *p* = 0.0332 vs. CTRL, #### *p* < 0.0001 vs. CTRL, **** *p* < 0.0001 vs. RBC, using one-way ANOVA with the Newman–Keuls post-hoc test.

**Figure 7 ijms-26-03953-f007:**
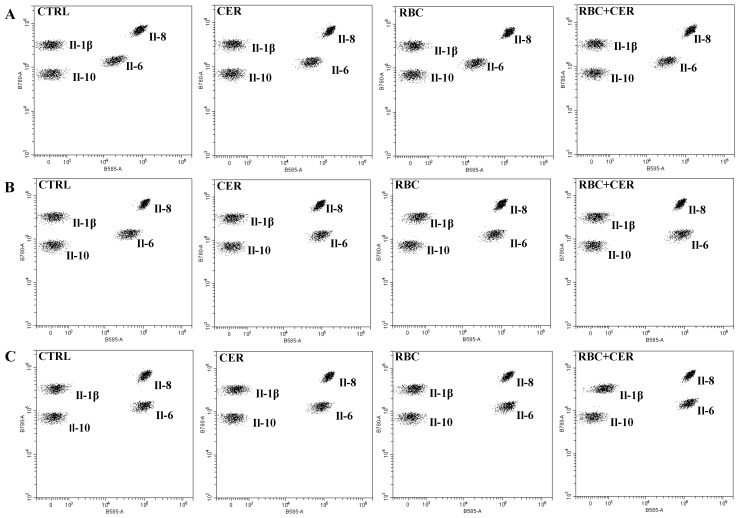
Effect of U87MG cell exposure to cerebrolysin (CER), RBC lysate (RBC), and RBC + CER on IL-1β, IL-6, IL-8, and IL-10 levels. Cytokines (IL-1β, IL-6, IL-8, and IL-10) were quantified by flow cytometry. Cells were treated with 125 µg/mL cerebrolysin, 10% RBC lysate, or both, while control cells were grown in standard medium. Cultures were maintained under (**A**) normoxia (37 °C, 95% humidity, 5% CO_2_), (**B**) continuous hypoxia (+200 mM CoCl_2_, 37 °C, 95% humidity, 5% CO_2_), or (**C**) cyclic hypoxia (1% O_2_ for 4 h → 21% O_2_ for 4 h → 1% O_2_ for 12 h →21% O_2_ for 4 h → 1% O_2_ for 4 h → 21% O_2_ for 4 h → 1% O_2_ for 12 h). Culture medium was collected after 48 h for cytokine measurements. Representative dot plots (beads of differing fluorescence intensity) depict PE signal intensities of IL-1β, IL-6, IL-8, and IL-10.

**Figure 8 ijms-26-03953-f008:**
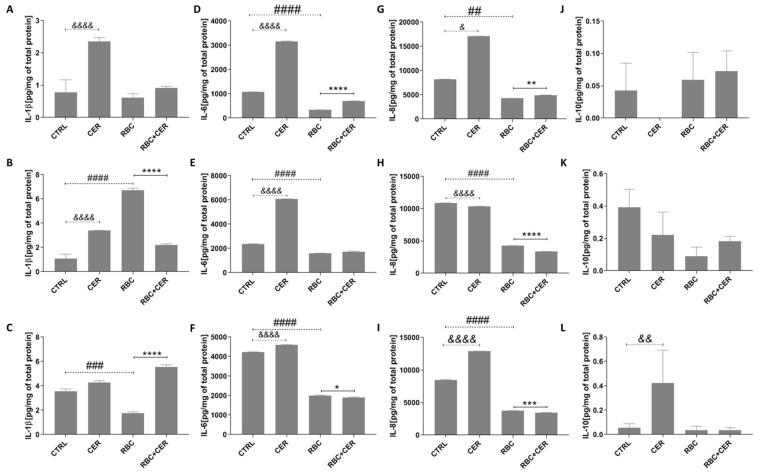
Effect of U87MG cell exposure to cerebrolysin (CER), RBC lysate (RBC), and RBC + CER on IL-1β, IL-6, IL-8, and IL-10 levels. Cytokine concentrations were measured by flow cytometry. Cells were treated with 125 µg/mL Cerebrolysin, 10% RBC lysate, or both (RBC + CER). Control cells were incubated in standard medium without additives. Cultures were maintained under (**A**,**D**,**G**,**J**) normoxia (37 °C, 95% humidity, 5% CO_2_), (**B**,**E**,**H**,**K**) continuous hypoxia (+200 mM CoCl_2_, 37 °C, 95% humidity, 5% CO_2_), or (**C**,**F**,**I**,**L**) cyclic hypoxia (1% O_2_ for 4 h → 21% O_2_ for 4 h → 1% O_2_ for 12 h → 21% O_2_ for 4 h → 1% O_2_ for 4 h → 21% O_2_ for 4 h → 1% O_2_ for 12 h). Culture medium was collected after 48 h for analysis. Data represent mean values ± SEM from three independent experiments (*n* = 3). & *p* < 0.05; && *p* < 0.001; &&&& *p* < 0.00001 CTRL vs. Cer; ## *p* < 0.01; ### *p* < 0.001; #### *p* < 0.0001 CTRL vs. RBC; * *p* < 0.05; ** *p* < 0.01; *** *p* < 0.001; **** *p* < 0.0001 RBC vs. CER, using one-way ANOVA with the Newman–Keuls post-hoc test.

**Figure 9 ijms-26-03953-f009:**
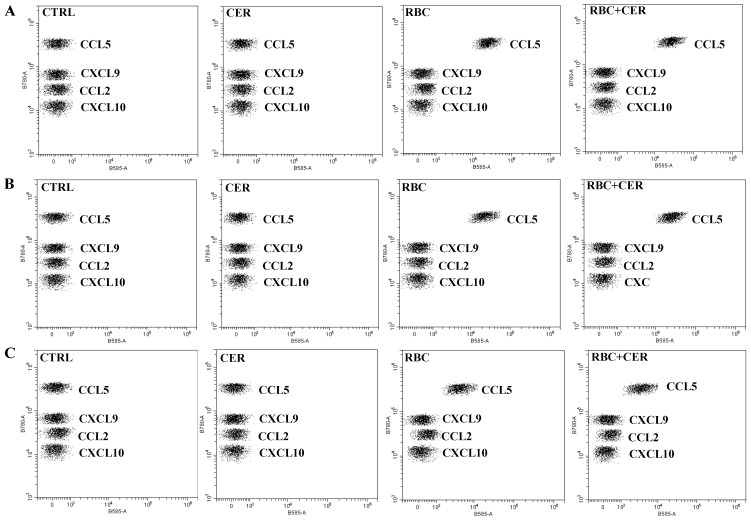
Effect of U87MG cell exposure to cerebrolysin (CER), RBC lysate (RBC), and RBC + CER on CCL5, CXCL9, CCL2, and CXCL10 levels. Chemokine concentrations were measured by flow cytometry. Cells were treated with 125 µg/mL cerebrolysin, 10% RBC lysate, or both (RBC + CER). Control cells were incubated in standard medium. Cultures were maintained under (**A**) normoxia (37 °C, 95% humidity, 5% CO_2_), (**B**) continuous hypoxia (+200 mM CoCl_2_, 37 °C, 95% humidity, 5% CO_2_), or (**C**) cyclic hypoxia (1% O_2_ for 4 h → 21% O_2_ for 4 h → 1% O_2_ for 12 h → 21% O_2_ for 4 h → 1% O_2_ for 4 h → 21% O_2_ for 4 h → 1% O_2_ for 12 h). Culture medium was collected after 48 h. Data represent mean values ± SEM from three independent experiments (*n* = 3).

**Figure 10 ijms-26-03953-f010:**
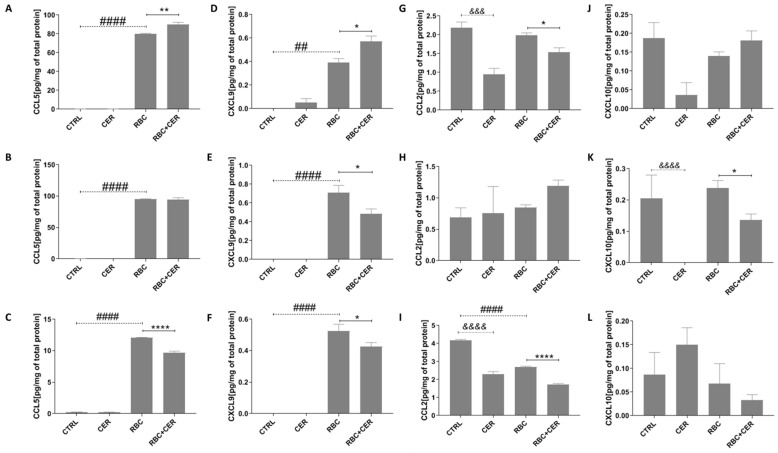
Effect of U87MG cell exposure to cerebrolysin (CER), RBC lysate (RBC), and RBC + CER on CCL5, CXCL9, CCL2, and CXCL10 levels. Chemokine concentrations were measured by flow cytometry. Cells were treated with 125 µg/mL cerebrolysin, 10% RBC lysate, or both (RBC + CER). Control cells were incubated in standard medium. Cultures were maintained under (**A**,**D**,**G**,**J**) normoxia (37 °C, 95% humidity, 5% CO_2_), (**B**,**E**,**H**,**K**) continuous hypoxia (+200 mM CoCl_2_), or (**C**,**F**,**I**,**L**) cyclic hypoxia (1% O_2_ for 4 h → 21% O_2_ for 4 h → 1% O_2_ for 12 h → etc.) for 48 h. Data represent mean values ± SEM from three independent experiments (*n* = 3). &&& *p* < 0.0001; &&&& *p* < 0.00001 CTRL vs. Cer; ## *p* < 0.01; #### *p* < 0.0001 CTRL vs. RBC; * *p* < 0.05; ** *p* < 0.01; **** *p* < 0.0001 RBC vs. CER, using one-way ANOVA with the Newman–Keuls post-hoc test.

**Table 1 ijms-26-03953-t001:** Immunofluorescence score for COX-1 protein expression in U87MG cells, based on staining intensity: negative; ± weak positive; + positive; ++ moderate. Scoring was performed independently by two researchers (KP and IBB) blinded to sample identity. A—normoxic, B—continuous hypoxic, C—cyclic hypoxic conditions.

	Ctrl	Cer	RBC	Cer + RBC
A	+	+	++	+
B	+	+	++	+
C	±	+	++	+

**Table 2 ijms-26-03953-t002:** Immunofluorescence score for COX-2 protein expression in U87MG cells, based on staining intensity: negative; ± weak positive; + positive; ++ moderate; +++ strong positive. Scoring was performed independently by two researchers (KP and IBB) blinded to sample identity. A—normoxic, B—continuous hypoxic, C—cyclic hypoxic conditions.

	Ctrl	Cer	RBC	Cer + RBC
A	+	+	+	+
B	+	+	+++	+
C	+	+	+++	±

## Data Availability

Data is contained within the article.

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
