# Peer review of "Influence of Exogenous Neuropeptides on the Astrocyte Response Under Conditions of Continuous and Cyclic Hypoxia and Red Blood Cell Lysate"

_ijms, 2025, doi:10.3390/ijms26093953_

Round 1

Reviewer 1 Report

Comments and Suggestions for Authors

The manuscript entitled The influence of exogenous neuropeptides on the response of astrocytes under conditions of continuous and cyclic hypoxia and red blood cell lysate was read with interest.

The study addresses a significant issue concerning the phenomenon of neuroinflammation. While neuroinflammation is a natural defence mechanism, its chronicity or excessive intensity has the potential to result in neuronal damage. Prolonged inflammation can lead to the degeneration of nerve cells, which is associated with numerous neurodegenerative diseases, including Alzheimer's disease, Parkinson's disease, and multiple sclerosis.

The following comments and suggestions are offered for consideration:
Firstly, the punctuation errors require correction.Secondly, the organisation of the entry for co-authors of the publication must be improved.Thirdly, the choice of cell line must be justified, explaining why this particular line was chosen and why other lines dedicated to neuro research would not be better.

There is an absence of information regarding cerebrolysin itself, and the justification for this work is unclear, given its utilisation as an adjuvant in the treatment of senile dementia of the Alzheimer's type and dementia of vascular origin. It is also used in post-traumatic deficits and craniocerebral trauma. It is imperative to ascertain whether preclinical studies have investigated the immunomodulatory effects of this substance.

A detailed description is required on the methodology employed to assess immunofluorescence levels, and the results thereof, particularly with regard to the expression of COX-1 and COX-2 proteins.

Author Response

Dear Reviewer,

Thank you very much for your time, comments and detailed revision. We will try to correct the manuscript, so it could meet your expectations. Below we present the answer to your suggestions.

  1. “Firstly, the punctuation errors require correction.” – corrected, as seen in attached file, we hope every error is corrected. If anything needs more corrections, please let us know.
  2. “Secondly, the organisation of the entry for co-authors of the publication must be improved.” – improved as seen in the attached file. If anything else should be rearranged, please let us know.
  3. “Thirdly, the choice of cell line must be justified, explaining why this particular line was chosen and why other lines dedicated to neuro research would not be better.” –

Human astrocytoma U-87 MG cells are widely used in scientific research. Since 2015, over 4,000 and almost 500 scientific papers using this cell line have been published (according to PubMed) in the last year alone. These are cells that are easy to maintain in culture. They do not have complicated requirements for culture. This glioma cell line is well characterized in terms of growth characteristics, morphology and gene expression, which makes it a valuable research tool. In addition, U-87 MG is a homogeneous cell line. Most cells in the population have the same genetic composition and therefore have similar characteristics. The cells express a variety of proteins, growth factors and/or their receptors, which are the targets of many studies. These cells are used to study cellular processes, drug screening and testing, in studies of chemokine and cytokine synthesis under conditions of toxic effects of various factors.

RSC Med Chem. 2025 Jan 20. doi: 10.1039/d4md00876f.; Comp Biochem Physiol C Toxicol Pharmacol. 2025 Jan;287:110065. doi: 10.1016/j.cbpc.2024.110065. Epub 2024 Nov 4.; Int J Mol Sci. 2022 Aug 1;23(15):8536. doi: 10.3390/ijms23158536. Neurotoxicology, 2017 May:60:82-91. doi: 10.1016/j.neuro.2017.03.001; Am J Cancer Res. 2021 Apr 15;11(4):1185-1205; Int J Mol Sci. 2022 Aug 1;23(15):8536. doi: 10.3390/ijms23158536.; Life Sci. 2015 Aug 1;134:16-21. doi: 10.1016/j.lfs.2015.04.024;

  1. “There is an absence of information regarding cerebrolysin itself, and the justification for this work is unclear, given its utilisation as an adjuvant in the treatment of senile dementia of the Alzheimer's type and dementia of vascular origin. It is also used in post-traumatic deficits and craniocerebral trauma. It is imperative to ascertain whether preclinical studies have investigated the immunomodulatory effects of this substance.”-

Thank you very much for this comment. We have emphasized the preclinical studies involving the immunomodulating properties of Cerebrolysin, included in the introduction and discussion (marked in green in the manuscript), (introduction : In vitro and in vivo research has shown that Cerebrolysin can reduce astrogliosis and enhance neurogenesis in rats subjected to experimental traumatic brain injury [24]. In neuronal cultures derived from the cerebral cortex, Cerebrolysin has been observed to improve glutamine transporter function and to increase the activity of glutamine pathways [25]. In experimental models, Cerebrolysin has also been found to diminish inflammatory responses [26]., discussion: Secondary inflammation processes plays a critical role in the pathogenesis of acute and chronic nervous system injury. Current research efforts focus on ways to modulate this process, since elucidating its underlying mechanisms may lead to new therapeutic strategies. Cerebrolysin, a mixture of low-molecular-weight neuropeptides and free amino acids, is one potential agent for regulating inflammation and promoting neuroprotection, neuroplasticity, and neuronal regeneration [27-28]. This brain-specific, pleiotropic neuroprotective compound has been proposed as a promising option for stroke and traumatic brain injury [26, 29-31). Previous studies show that Cerebrolysin acts through multiple molecular pathways to enhance functional recovery following neurological diseases and injuries, including interactions with GluR1 and GABA RA/B receptors, regulation of intracellular signaling cascades (PI3K/Akt, NF-κB, JNK, and p38-MAPK), and alterations in molecular mediators relevant to trophic activity (NGF, BDNF, IGF-1), glucose transport (GLUT1), inflammation (TNF-α, IL-1β), and neurotransmission (cholinergic, glutamatergic, and GABAergic) [ 25-26, 32-34]. Cerebrolysin’s neuroprotective and neurotrophic properties have been confirmed both in vitro and in animal models of ischemic stroke, as well as in clinical studies involving ischemic stroke patients and those with traumatic brain injury diagnosis [25-26,31,35-36). These properties encompass anti-apoptotic activity, mitigation of glutamate excitotoxicity, reduction of free oxygen radicals, and modulation of microglial activation and neuroinflammatory [25, 29-31, 34, 37-43 ].In stroke, reactive astrocytes commonly release pro-inflammatory cytokines that can exacerbate neuronal damage. Cerebrolysin can moderate this inflammatory response. Some studies indicate that it significantly improves functional recovery in rat models of mild traumatic brain injury by reducing astrogliosis and axonal damage, enhancing neurogenesis, and lowering blood–brain barrier (BBB) disruption [31]. In a glutamate-toxicity model, Cerebrolysin also decreases IL-1β levels and raises IL-10 levels, thereby mitigating glutamate-mediated neuroinflammation [25]. IL-10 plays a critical protective role by inhibiting pro-inflammatory cytokine release, thus helping suppress both immunoproliferative and inflammatory processes [44]. Because an imbalance between pro-inflammatory and anti-inflammatory cytokines can significantly affect stroke outcomes, an IL-10 increase driven by Cerebrolysin may foster a more balanced immune response and support tissue repair. This mechanism may also limit excessive inflammation and protect brain tissue in injury contexts. After traumatic brain injury (TBI), Cerebrolysin enhances neurological scores, lowers cerebral edema, improves BBB permeability, and reduces inflammatory cytokines such as TNF-α, IL-1β, IL-6, and NF-κB [26]. Parallel findings have been reported in TBI patients, where Cerebrolysin decreases TNF-α, IL-1β, IL-6, and NF-κB. Moreover, Cerebrolysin can downregulate inflammatory responses in intracranial hemorrhage. In murine intracerebral hemorrhage models, Cerebrolysin-treated animals exhibited significant reductions in IL‑1β, IL‑6, and TNF‑α, and in aquaporin-4 (AQP4) – a main mediator of vasogenic edema. A decrease in cytotoxic astrocyte edema was also noted [45].)

Concerning the Cerebrolysins’ in vivo and in vitro studies it seemed reasonable to assume that it could be used with benefit in patients with acute brain injury, aswell as with chronic. In the metaanalysis however and single studies the efficacy of Cerebrolysin used in TBI and stroke is better than in SAH. As stated in the manuscript : “A central question is why some clinical results on Cerebrolysin use in TBI, SAH, or stroke show less robust statistical significance than those seen in TBI alone.” We tried to answer it in our investigation with the red blood cell lysate imitating the haemorhhage.

  1. “A detailed description is required on the methodology employed to assess immunofluorescence levels, and the results thereof, particularly with regard to the expression of COX-1 and COX-2 proteins.”

Thank you very much for that comment. We’ve tried to add the methodology to the 2.3.1 chapter as follows:

“. Imaging of COX-1and COX-2 expression

COX-1 and COX-2 levels were detected by fluorescence microscopy. U87MG cells (5 × 105) were treated with 125 µg/ml Cerebrolysin, 10% RBC lysate, or both (CER+RBC). Control cells were cultured in standard medium without additives. Incubations were under normoxic (37°C, 95% humidity, 5% COâ‚‚), continuous hypoxic (+200 mM CoClâ‚‚), or cyclic hypoxic conditions (1% Oâ‚‚ for 4 h → 21% Oâ‚‚ for 4 h → 1% Oâ‚‚ for 12 h → etc.) for 48 h.  Following incubation, cells were washed in PBS and used for confocal microscopy to visualize COX-1 and COX-2 proteins. Cells were cultured on glass coverslips in six-well plates under standard conditions, washed with PBS, fixed for 15 minutes at room temperature with 4% buffered formalin, and then washed again. Permeabilization was done with 0.5% Triton X-100 in PBS. After additional washing, cells were incubated for 1 hour at room temperature with mouse anti-COX-1 and anti-COX-2 antibodies (1:50; Santa Cruz Biotechnology, Dallas, TX, USA). They were then rinsed and further incubated for 45 minutes at room temperature with anti-mouse IgG FITC-conjugated secondary antibodies (1:80; Sigma-Aldrich, PoznaÅ„, Poland). After another wash in PBS, the nuclei were stained with DAPI for 15 minutes at room temperature. Samples were examined on an Olympus FV1000 confocal microscope (IX81 inverted platform), using two-channel acquisition and sequential scanning to separately detect DAPI and FITC fluorescence. Enzyme expression was visible as green fluorescence.  Merged fluorescence images were then prepared. Experiments were performed as six separate assays for each COX (each assay in three replicates).”

Thank you for your suggestions, please let us know if there is anything else that we could improve in our manuscript.

Kind regards,

Klaudyna Kojder

Reviewer 2 Report

Comments and Suggestions for Authors

In the article entitled “The impact of exogenous neuropeptides on the astrocyte response under conditions of continuous and cyclic hypoxia and red blood cell lysate”.

The authors do a study investigating the impact of exogenous neuropeptides (Cerebrolysin) on the astrocyte response under conditions of continuous and cyclic hypoxia and red blood cell lysate exposure. COX-1 and COX-2 expression, as well as levels of inflammatory cytokines and chemokines, were assessed using fluorescence microscopy and flow cytometry.

Some points need to be addressed that could improve this important study.

-Expand the experimental model: Include primary astrocytes derived from human or animal tissue to validate the results in a more physiological context.

-Incorporate co-cultures with neurons or microglia to assess cellular interactions during inflammation and response to Cerebrolysin.

-Evaluate functional effects: Perform cell viability assays (MTT, apoptosis, autophagy) to determine if reduction of inflammatory mediators translates into tangible cell protection.

-Analyze the integrity of the blood-brain barrier in vitro to assess whether Cerebrolysin could preserve its functionality in the face of hypoxic damage.

-Explore molecular mechanisms: including analysis of key signaling pathways, such as NF-κB, MAPK or PI3K/Akt, to understand how Cerebrolysin modulates the inflammatory response.

-Evaluate changes in gene expression by quantitative PCR or RNA-seq for a deeper understanding of the effects of the peptide.

Overall.

This study provides valuable information on the impact of Cerebrolysin in modulating the inflammatory response in astrocytes under hypoxia and exposure to red blood cell lysate. Their findings suggest that the peptide may have a neuroprotective role, although further studies are required to validate its efficacy and better understand its mechanisms of action.

To strengthen the impact of the study, it would be advisable to incorporate physiological models, explore cellular functionality and analyze key molecular pathways.

Minor points.

The scale bar in Figures 1, 2, 4 and 5 has not been added.

Adjust the figures so that they are not disintegrated throughout the manuscript.

Author Response

Dear Reviewer,

We are very grateful for all the comments and suggestions for new research. We agree  that the results of new experiments would enrich our manuscript. However, the topic we have undertaken in the current research did not assume such a wide spectrum. Their implementation would require very large financial outlays and an extended research time, which would involve writing a new research project, obtaining funds for such extensive research. This is undoubtedly the direction of our further research, but at the moment we cannot undertake it on such a wide scale.

The minor points included in the revier: the figures were adjusted, we hope it is not desintagrated now, and the scale bars were added in the figures  1,2, 4,5 as you’ve suggested.

Please let us know, if we can improve our manuscript in any other way.

Kinds regards,

Klaudyna Kojder

Round 2

Reviewer 2 Report

Comments and Suggestions for Authors

Thanks to the authors for their replies. I consider that with the changes made, the manuscript improved substantially.